# Extremum Seeking with Surrogate Gradients: Scalable Derivative-Free Optimization for High-Dimensional Black-Box Functions

## Abstract

Derivative-free optimization remains a central challenge in high-dimensional black-box problems where gradients are unavailable and evaluations are moderately expensive or must be performed under tight, non-parallelizable time budgets. Classical approaches such as Bayesian optimization suffer computational cost from model training and acquisition-function optimization, while many evolutionary strategies can be sample-inefficient. Classical Zeroth-order methods are also often data-inefficient requiring evaluation queries for gradient estimation. We propose a hybrid framework that combines Extremum Seeking (ES) with surrogate-gradient (SG): a Gaussian process surrogate is trained on data collected during the optimization process and used to predict approximate local gradients at the current iterate, while ES provides structured local perturbative exploration via sinusoidal dithering. By using predicted gradients to form direct update directions, the method avoids a separate acquisition-function optimization step and reduces per-iteration overhead in sequential deployments. To improve robustness and adaptive learning rate we project surrogate gradients according to their posterior uncertainty and apply Adam on the surrogate gradient. On high-dimensional synthetic control benchmarks, our approach outperforms standard ES and comparable to the trust-region BO on the tasks studied.

## 1 Introduction

High-dimensional black-box optimization is a critical task in numerous scientific and engineering domains, ranging from hyperparameter tuning of large neural networks to the online control of complex physical systems. Despite the "curse of dimensionality," training neural networks with millions and billions of parameters has become feasible. Beyond GPU computing, the primary enabler is automatic differentiation combined with gradient descent, which scales gracefully with dimensionality (Paszke et al., 2017). However, many practical optimization problems are characterized by objective functions where gradients are inaccessible. We are particularly interested in time-constrained and moderately costly problems, where evaluating the objective function on a physical system takes only a few seconds. Such problems arise across numerous scientific and engineering domains involving machine control, real-time planning, or design. Specific examples include autonomous vehicle path planning in dense traffic (Paden et al., 2016; Chen et al., 2019), particle accelerator beam tuning (Hwang et al., 2022; 2024; et. al., 2024), and rover trajectory planning in unstructured terrain (Eriksson et al., 2019).

In such settings, where gradient information is inaccessible, derivative-free optimization (DFO) provides a natural alternative. Bayesian Optimization (BO) is highly sample-efficient but struggles to scale beyond a few dozen dimensions due to the computational overhead of building and optimizing a global surrogate model, which can become a bottleneck for real-time applications. Zeroth-order methods, which approximate gradients using finite differences, often require an excessive number of function queries and are sensitive to noise (Liu et al., 2020; **?**). Several approaches have been proposed to mitigate this query burden, such as efficient sampling policies (Zhai et al., 2024) or strategic active learning (Müller et al., 2021). Nevertheless, the need for function queries in gradient estimation persists, and active learning itself can be computationally expensive for real-time optimization tasks. To address this, (Shu et al., 2023) introduces GP surrogate modeling for gradient

estimation without additional queries, relying solely on training data from the optimization trajectory. However, in the absence of exploration, the resulting gradient estimates may be less reliable. The uncertainty of gradient estimation is addressed in (Nguyen et al., 2022) by estimating the most probable descent direction based on uncertainty of gradient predictions and maximizing probability of descent via active learning; however, this still incurs a querying step.

To address these challenges, we propose a hybrid framework that synergistically combines Extremum Seeking (ES) (Scheinker, 2013; Scheinker & Scheinker, 2016) and Surrogate Gradient (SG) ascent for objective function maximization. ES is a real-time, model-free optimization technique from control theory that introduces sinusoidal perturbations (dithers) to the system inputs to perform gradient ascent on average, scaling gracefully with dimensionality. This structured exploration provides a continuous stream of data around the optimization trajectory in control space. We leverage this data to train a local Gaussian Process (GP) model, which acts as a surrogate for the objective function. Crucially, instead of modeling the entire high-dimensional space, we only require the surrogate to provide a reliable local gradient—the Surrogate Gradient (SG) near the optimization path.

Our contributions are as follows:

- We introduce a hybrid ES+SG framework where ES provides exploration while performing gradient ascent on average, and SG provides further acceleration.
- We enhance this framework by estimating gradient at the mid point between data, incorporating the Adam optimizer (Kingma & Ba, 2014) for adaptive acceleration and an uncertainty-aware gradient projection scheme. This scheme uses the GP's predictive uncertainty to suppress the control parameter update step toward unreliable gradient direction (Nguyen et al., 2022), improving stability and convergence speed.
- (?) We provide a theoretical analysis that frames ES as an efficient sampling strategy for a local surrogate model and discuss the convergence properties of our algorithm in the presence of inexact gradients.
- We demonstrate the effectiveness of our approach on high-dimensional synthetic benchmarks, showing performance gains over standalone ES. The benchmark compares with TurBO (Eriksson et al., 2019) and SG ascent without ES similar to ZoRD (Shu et al., 2023).

## 2 RELATED WORK

Our work lies at the intersection of zeroth-order optimization, Bayesian optimization, and control theory.

**Zeroth-Order (ZO) Optimization.** ZO methods optimize black-box functions by estimating gradients from function evaluations. Classic techniques include finite-difference methods and Simultaneous Perturbation Stochastic Approximation (SPSA) (Spall, 1992). More recent approaches use randomized smoothing to create a differentiable surrogate (Chen, 2023). Another recent work (Shu et al., 2023) leverage the optimization trajectory to build a model for the gradient, similar in spirit to our work.

**Bayesian Optimization (BO).** BO is a widely used DFO method valued for its sample efficiency, achieved by constructing a probabilistic surrogate model (typically a GP) of the objective function and leveraging an acquisition function to guide the search. However, modeling the global landscape of the objective requires data that grows exponentially with dimensionality. To mitigate this, local BO methods restrict to a trust region around the current best solutions Eriksson et al. (2019) or estimate gradient from surrogate model at the current point - similar to ZO methods. For example, Nguyen et al. (2022) introduced an algorithm that iteratively identifies the most probable descent direction based on a GP model. Our work also employs a GP for gradient information but differs in its exploration strategy: whereas their method alternates between a learning phase and an optimization step, our ES+SG framework integrates exploration and exploitation simultaneously.

**Extremum Seeking (ES).** ES is a classical adaptive control technique for real-time, model-free optimization (Scheinker, 2013). It uses periodic perturbation signals to estimate the gradient of an unknown input-output map and drive it to an extremum. Its key advantages are its simplicity, model-free nature, and proven robustness to noise and system dynamics. It has been successfully

applied in various engineering fields, but its integration with modern machine learning techniques for accelerated convergence is a recent development.

# 3 THE ES+SG FRAMEWORK

Our proposed method, Extremum Seeking with Surrogate Gradients (ES+SG), is a hybrid algorithm designed for efficient, high-dimensional DFO. It couples the exploratory power of ES with the sample-efficient exploitation of gradient prediction from a local surrogate model.

## 3.1 EXTREMUM SEEKING

Assume that $f : \mathbb{R}^D \to \mathbb{R}$ is the second order continuous Lipschitz. We adopt a bounded-update-rate ES scheme that perturbs each component of the control vector $\boldsymbol{x} \in \mathbb{R}^D$ with a high-frequency dither:

$$\dot{x}_i = \sqrt{\alpha_i \omega_i} \cos\left(\omega_i t \mp \kappa f(\boldsymbol{x})\right), \quad i = 1, \ldots, D, \tag{1}$$

where $\boldsymbol{\alpha} \in \mathbb{R}_{>0}^D$ and $\boldsymbol{\omega} \in \mathbb{R}_{>0}^D$ are the dither amplitudes and frequencies respectively, and $\kappa > 0$ is the gain parameter. As shown in (Krstić & Wang, 2000; Dürr et al., 2013; Dürr & Ebenbauer, 2017; Ariyur & Krstić, 2003; Scheinker & Krstić, 2014), under standard ES conditions such as differentiability of $f(\boldsymbol{x})$ and pairwise-distinct $\omega_i = \omega r_i$ with $r_i \neq r_j$ for $i \neq j$, in the limit as $\omega$ is increased towards $\infty$ an averaging analysis yields averaged gradient dynamics

$$\dot{\bar{\boldsymbol{x}}} = \pm \frac{\kappa}{2} \boldsymbol{\alpha} \, \nabla_{\boldsymbol{x}} f(\bar{\boldsymbol{x}}), \tag{2}$$

where $\omega$ must be chosen sufficiently large relative to $\kappa$ and also a bound on $|f(\boldsymbol{x})|$ for $\mathbf{x} \in K$ for some compact set $K \subset \mathbb{R}^D$. This result holds for *arbitrary* finite dimension $D$, highlighting the *scalability* of ES to high-dimensional problems, as experimentally demonstrated in Scheinker et al. (2019) to simultaneously tune $> 100$ magnets of a particle accelerator for real-time optimization. This robust method has also been studied for Tokamak stabilization Dubbioso et al. (2025).

For a discrete-time implementation with sampling times $t \in \mathbb{Z}$, we use the following ES step,

$$\Delta \boldsymbol{x}_t^{\text{ES}} = \sqrt{\boldsymbol{\alpha}\boldsymbol{\omega}} \sin\left(\boldsymbol{\omega} t \mp \kappa \, f(\boldsymbol{x}_t)\right), \tag{3}$$

based on a finite-difference approximation of the continuous-time dynamics in Equation 1. See Appendix A.

Despite its theoretical scalability, applying ES in high dimensions presents practical challenges. As dimensionality $D$ grows, ensuring adequate separation between the dither frequencies (or their fractional tunes, $\boldsymbol{\nu} = \boldsymbol{\omega}/2\pi$) becomes increasingly difficult, which can slow down the ES convergence process. Furthermore, the optimal selection of hyper-parameters, especially the gain $\kappa$, is challenging without a priori knowledge of the function's landscape (Scheinker et al., 2022). In order to overcome these challenges, we exploit surrogate modeling.

## 3.2 SURROGATE GRADIENT (SG) STEP

ES not only perform gradient ascent (or descent) on average, but also introduce exploration around the optimization path. The sinusoidal dithering of distant frequencies enables structured exploration over well-separated data over the control domain, making it suitable for surrogate modeling along the optimization trajectory. For example, for $\kappa = 0$, the trajectory of each parameter in Equation 1 can be analytically solved as

$$x_i(t) = x_i(0) + \sqrt{\frac{\alpha_i}{\omega_i}} \sin(\omega_i t), \tag{4}$$

with each parameter exploring a region of radius $\sqrt{\alpha_i/\omega_i}$ about its current position. During optimization, for $\kappa \neq 0$, the analysis in (Scheinker, 2013; Scheinker & Scheinker, 2016) shows that for $\omega$ sufficiently large such exploration continues to take place with radius proportional to $\sqrt{\alpha_i/\omega_i}$ about the averaged trajectory. We employ the data $\mathcal{D}_t = \{(\boldsymbol{x}_i, f(\boldsymbol{x}_i))\}_{i=1}^t$ on the ES optimization path to train a Gaussian Process (GP) surrogate.

It is also intuitive that ES perturbations would reduce model uncertainty in the limit of small learning rates. Without perturbation, data collected strictly along the gradient path would densely populate

a line but fail to explore directions orthogonal to it. By contrast, ES-induced dithering expands the trajectory into a small volume around the path, enriching the dataset with more diverse samples. This additional variation would improve the surrogate's ability to capture local curvature and gradient direction. Figure 1 demonstrates this effect for a moderately small learning rate on the 2D Goldstein–Price function (Picheny et al., 2013). Figure 2 also support the idea showing much better cosine similarities between predicted and true gradient direction over ES perturbed trajectory.

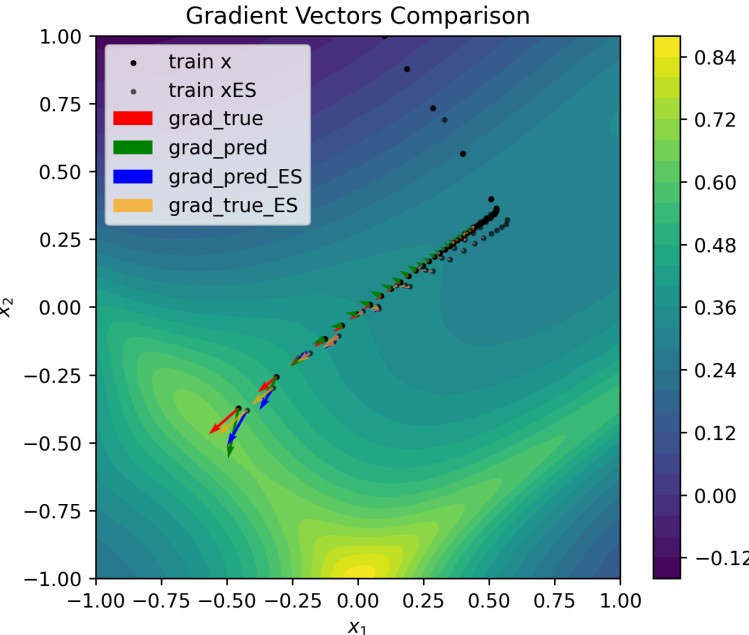

Figure 1: Gradient prediction versus ground truth for two datasets: (i) samples strictly along the gradient path with a fixed learning rate, and (ii) ES-perturbed samples around the same path. Perturbations lead to better alignment with the true gradient, as indicated by the improved agreement between predicted (green/blue) and true (red/yellow) arrows.

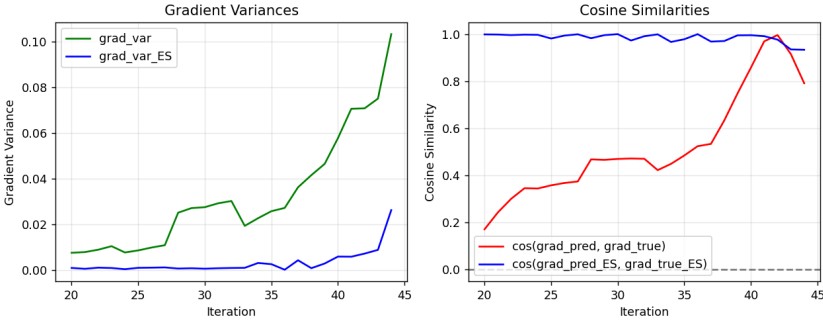

Figure 2: Comparison of gradient quality metrics: trace of the Jacobian variance (grad_var) and cosine similarity between predicted and true gradients. ES perturbations yield consistently higher scores, reflecting improved surrogate accuracy.

The combined update rule integrates the ES step with the surrogate gradient (SG) step:

$$x_{t+1} = x_t + \eta_{\text{SG}} \Delta x^{\text{SG}} + \eta_{\text{ES}} \Delta x^{\text{ES}} \tag{5}$$

where $\eta_{\text{SG}}$ is learning rates for the SG step. Since the effective learning rate of ES is governed by Equation 2, We set $\eta_{\text{ES}} = 1$ by default, but retain it as a scheduling parameter to gradually reduce the ES step size as the optimizer approaches the optimum.

Figure 3 benchmarks the hybrid method on the 200-dimensional normalized Rastrigin function, comparing ES, SG, and their combination. The ES+SG strategy achieves higher objective values and better gradient alignment.

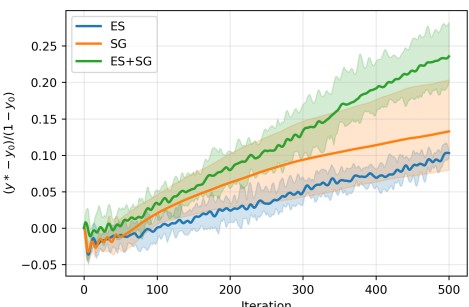 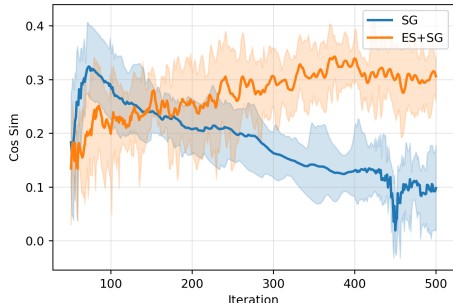

Figure 3: Performance comparison on the 200-dimensional Rastrigin function using ES, SG, and their combination (ES+SG). Left: normalized objective function value, with $y_0$ denoting the initial value and $y*$ the global optimum. Right: cosine similarity between predicted and true gradients. Shaded regions indicate the 10% and 90% quantiles across 10 random restart samples, while solid lines show the mean.

When the evaluation of the physical objective function takes several seconds, we train the surrogate on $\mathcal{D}_{t-1}$ rather than $\mathcal{D}_t$, enabling asynchronous computation without delaying real-time optimization. Similarly, all the following discussion can be turn to single asynchronous approach to tightly exploit real time by $t \to t - 1$. Unlike asynchronous Bayesian Optimization (BO), gradient ascent (or descent) setting eliminates the need to penalize or "fantasize" the pending evaluation (González et al., 2016; Snoek et al., 2012), making single asynchronous approach relatively more trivial.

To reduce model training time, we truncated the training data to fewer than 400 iterations. This choice is justified because the prediction target is inherently local, while earlier trajectory points are typically far from the current optimization region and thus less informative.

### 3.2.1 ANCHORING GRADIENT EVALUATION POINTS AT MID POINTS OF OPTIMIZATION TRAJECTORY

We define SG step at the current point $\boldsymbol{x}_t$ by averaging gradient of GP posterior mean over the midpoints between $\boldsymbol{x}_t$ and the most recent $n_{\text{ave}}$ training points (we used, $n_{\text{ave}} = 4$):

$$\Delta \boldsymbol{x}_t^{\text{SG}} = \mathbb{E}_{\Omega_t} \left[ \nabla_{\boldsymbol{x}} \mu(\boldsymbol{x}) \right]$$

where $\mu(\boldsymbol{x})$ denotes the GP posterior mean and

$$\Omega_t = \left\{ \frac{\boldsymbol{x}_i + \boldsymbol{x}_t}{2} \middle| i \in \{t, t-1, \dots, t - n_{\text{ave}}\} \right\}. \tag{6}$$

This midpoint-based averaging addresses a limitation of GP with stationary kernels including Radial Basis Function (RBF): beyond the region covered by training data, posterior predictions tend to revert toward the prior mean, thereby biasing gradients toward the prior (see Fig. 4). Since recent evaluations are more likely to correspond to higher objective values (for maximization problem), anchoring gradient evaluations at midpoints along the recent trajectory reduces this bias and ensures that the surrogate gradient more accurately reflects the locally relevant portion of the optimization path. On top of that it is intuitive to estimate gradient at the mid-point of data in the finite difference (FD) sense: $\nabla_\epsilon f(x) \simeq \left( f(x + \epsilon/2) - f(x - \epsilon/2) \right) / \epsilon$

Indeed, it can be shown that the uncertainty of gradient is reduced between observations under some minor assumptions on the GP kernel. The Jacobian's variance $\boldsymbol{\Sigma_g}$ at $\boldsymbol{x}$ is (Müller et al., 2021;

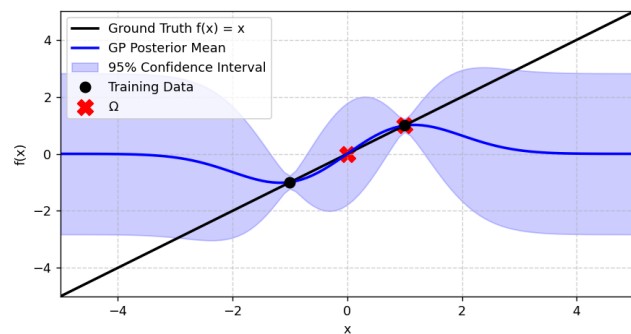

Figure 4: GP regression with an RBF kernel illustrating the tendency of the posterior mean (blue) to revert toward the prior mean outside the training data region (black dots). The true function $f(x) = x$ (black line) grows linearly, but the GP flattens toward the prior. Midpoint gradient evaluations ($\Omega$, red crosses) mitigate this bias by anchoring estimates to the recent optimization trajectory.

Nguyen et al., 2022):

$$\Sigma_g(\boldsymbol{x}) = \nabla K(\boldsymbol{x}, \boldsymbol{x})\nabla^\top - \nabla K(\boldsymbol{x}, \boldsymbol{X})\left(K(\boldsymbol{X}, \boldsymbol{X}) + \sigma^2 I\right)^{-1} K(\boldsymbol{X}, \boldsymbol{x})\nabla^\top, \tag{7}$$

where $\boldsymbol{X}$ denotes the training inputs, $K(\cdot, \cdot)$ is the GP kernel, and $\nabla$ on the left and right side of kernel represent the derivative with respect to its first and second input respectively.

**Theorem. 1.** Consider GP posterior conditioned on two observations at $\boldsymbol{X} = \{-\boldsymbol{v}, \boldsymbol{v}\}$, where $\boldsymbol{v} \in \mathbb{R}^D$ is an arbitrary vector in input domain. With a homoskedastic noise variance $\sigma^2$ and a stationary and isotropic kernel $K(\boldsymbol{x}, \boldsymbol{x}') = \phi(||\boldsymbol{x} - \boldsymbol{x}'||)$ that satisfies (i) $\phi \in C^2([0, \infty))$ with $\phi'(0) = 0$ and $\phi''(0)$ finite (so gradient variance exists), (ii) $\phi(r)$ is monotonically decreasing on $(0, \infty)$, and (iii) $|\phi'(r)|$ is nonincreasing on $(0, \infty)$, the following holds

$$\det \Sigma_g(0) < \det \Sigma_g(\pm\boldsymbol{v}) \tag{8}$$

for any $\boldsymbol{v} \neq 0$.

The proof is on Appendix B. Note that we define the criterion using the determinant of the Jacobian covariance matrix, rather than its trace as in the GiBO acquisition function (Müller et al., 2021). The trace corresponds to the sum of marginal variances across dimensions, whereas the determinant measures the hyper-volume of the uncertainty ellipsoid: the uncertainty region spanned by the gradient distribution.

### 3.2.2 ADAM OPTIMIZER WITH SG AND LEARNING RATE DECAY (ADAMSG)

Since we have estimated gradient, we can also apply the Adam optimizer (Kingma & Ba, 2014) to the surrogate gradient term $\Delta\boldsymbol{x}^{\text{SG}}$. This helps accelerate progress along consistent gradient directions and escape local optima. However, its momentum can also cause the optimizer to overshoot the optimum.

To counteract this, we introduce a learning rate decay schedule that adapts based on the objective function value rather than a fixed number of iterations. The learning rates are updated at each time step $t$ following logistic decay function,

$$\eta_t = \frac{\eta_0}{1 + \exp\left(\gamma\left(f(\boldsymbol{x}_t) - f_{\text{ref}}\right)\right)} \tag{9}$$

where $\gamma > 0$ is controls the steepness of the decay and $f_{\text{ref}}$ is a reference objective value. As the measured performance $f(\boldsymbol{x}_t)$ approaches this reference value, the learning rates are automatically reduced, promoting a finer-grained search as the optimizer converges. In our practice, we normalize objective function such that the optimum value is 1 while variance is about 1 so that $f_{\text{ref}}$ can be fixed around 0.8.

Figure 5 shows our typical choice of learning rate decay. We choose to decay $\eta_{ES}$ faster than $\eta_{SG}$ as SG step performs exploitation while simultaneous ES step performs exploration.

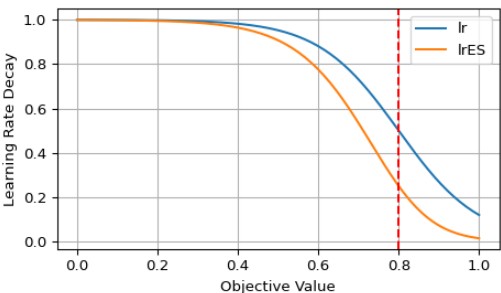

Figure 5: Learning rate decay as a function of objective function value.

### 3.2.3 UNCERTAINTY-AWARE SURROGATE GRADIENT STEP (uSG)

In high-dimensional settings with limited data, surrogate gradients can become unreliable. To address this, we leverage the GP's ability to quantify predictive uncertainty. Our approach, described below, provides a principled way to incorporate gradient uncertainty into the optimization process. Although we did not observe performance improvements on our benchmarks for this uncertainty awareness (next section), we consider this result informative and believe further investigation is necessary to understand the underlying reasons.

The uncertainty of the surrogate gradient $\Sigma_g$ is analytically derivable from the GP kernel as Equation 7. The uncertain gradient directions can be suppressed by multiplying the inverse of the uncertainty $\Sigma_g$ on the gradient of the GP mean $\nabla\mu$.

$$\boldsymbol{v}^* = \boldsymbol{\Sigma}_g^{-1}\nabla\mu$$

Nguyen et al. (Nguyen et al., 2022) showed that $\boldsymbol{v}^*$ corresponds to the most probable ascent direction. They also proved that the probability of ascent along a general direction $\boldsymbol{v}$ is explicitly given by

$$P = \Phi\left(\sqrt{\nabla\mu^T\boldsymbol{\Sigma}_g^{-1}\nabla\mu}\right), \tag{10}$$

where $\Phi(\cdot)$ is the cumulative distribution function of the standard normal distribution.

Although $\boldsymbol{v}^*$ identifies the most probable ascent direction, multiplying by $\boldsymbol{\Sigma}_g^{-1}$ distorts the magnitude information of the gradient. To address this, we project the GP mean gradient $\nabla\mu$ onto the unit vector of $\boldsymbol{v}^*$,

$$\hat{\boldsymbol{v}}^* = \frac{\boldsymbol{v}^*}{|\boldsymbol{v}^*|_2}, \tag{11}$$

$\hat{\boldsymbol{v}}^* = \boldsymbol{v}^*/\|\boldsymbol{v}^*\|_2$. In addition, we scale it down by the probability of ascent $P$. This yields the uncertainty-aware surrogate gradient step (uSG):

$$\Delta\boldsymbol{x}^{\text{uSG}} = P\,\hat{\boldsymbol{v}}^*\,(\nabla\mu\cdot\hat{\boldsymbol{v}}^*). \tag{12}$$

## 4 BENCHMARK

We evaluate our methods on high-dimensional synthetic optimization problems. For all benchmarks, ES is applied for the first few iterations to collect initialization data. The following tags are used to distinguish the optimizers:

- **ES**: Extremum Seeking only
- **SG**: Surrogate Gradient ascent without ES
- **ES+SG**: Hybrid method combining ES and SG

- **ES+SGadam**: Hybrid method with Adam optimization applied to SG
- **ES+uSGadam**: Hybrid method with Adam optimization applied to the uncertainty-aware surrogate gradient
- **TurBO**: Trust Region Bayesian Optimization (Eriksson et al., 2019), with success and failure tolerance counts set to 3 for expanding or shrinking the search region by a factor of two.

Figures 6, 7, and 8 report performance on the Rosenbrock (Rosenbrock, 1960), Rastrigin (Rastrigin, 1974), and Ackley (Ackley, 1987) test functions, respectively. Each optimizer is initialized from the same set of 10 random starting points, and objective values are normalized by their corresponding initial values. Shaded regions indicate the 10%–90% quantile range, while solid lines show the mean of the best-so-far objective value at each iteration across trials.

Overall, **ES+SGadam** outperforms other methods, including the state-of-the-art TurBO, with the performance gap widening in higher dimensions. Once the surrogate model is trained from a few ES steps, Adam-based methods exhibit rapid progress. In contrast, the uncertainty-aware variant (**ES+uSGadam**) underperformed unexpectedly, suggesting further study. Importantly, all SG variants remain efficient—under 5 seconds per iteration for 200D problems—making the approach practical for moderately expensive objectives while avoiding the acquisition-function overhead of BO methods.

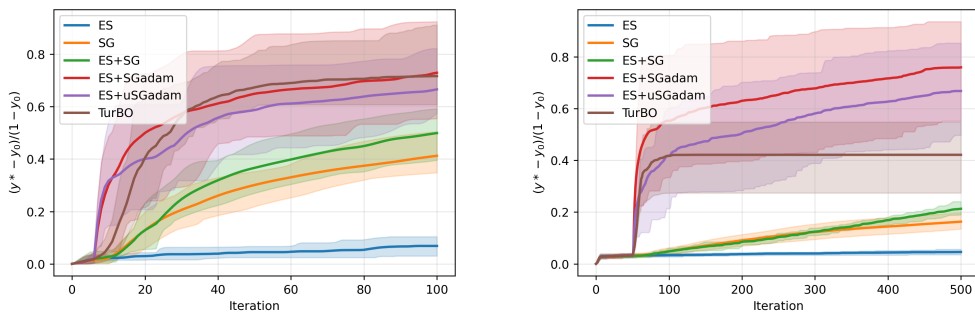

Figure 6: Benchmark on Rosenbrock function. Left: 20D, Right:200D.

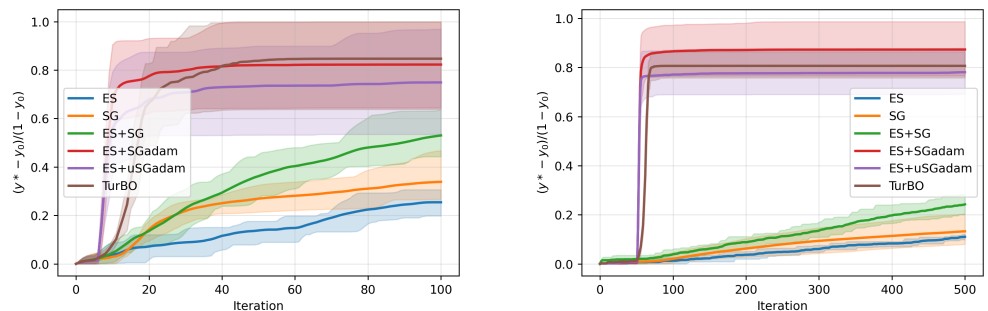

Figure 7: Benchmark on Rastrigin function. Left: 20D, Right:200D.

## 5 CONCLUSION

We introduced ES+SG, a hybrid framework that combines Extremum Seeking (ES) with surrogate gradient (SG) estimation to address high-dimensional derivative-free optimization. By leveraging ES-induced structured perturbations for exploration and a Gaussian process surrogate for local gradient prediction, the method balances scalability with sample efficiency. Our theoretical analysis

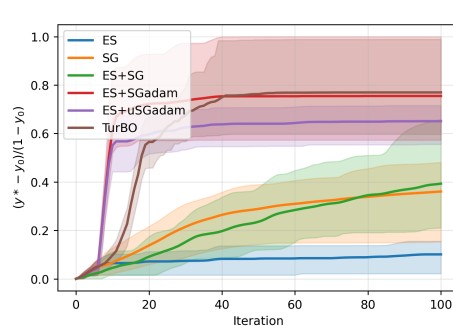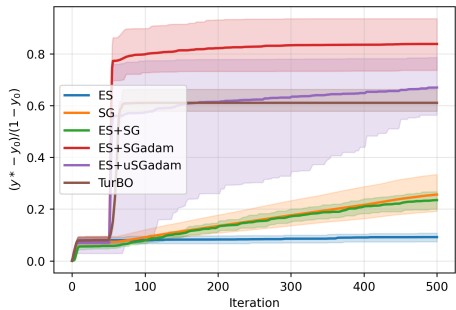

Figure 8: Benchmark on Ackley function. Left: 20D, Right:200D.

shows that ES exploration enriches the surrogate model with informative data, reducing gradient uncertainty, while midpoint anchoring further alleviates bias in GP predictions. The integration of adaptive learning (Adam) and uncertainty-aware projection schemes extends the robustness of the framework.

On high-dimensional synthetic benchmarks, ES+SG with Adam outperformed both standalone ES and competitive trust-region BO methods, while maintaining low per-iteration computational overhead suitable for real-time applications. Interestingly, our experiments revealed that the uncertainty-aware variant did not improve performance, pointing to a promising direction for further investigation into the interaction between gradient reliability and optimization dynamics.

Overall, ES+SG offers a scalable, practical alternative for real-time, moderately expensive high-dimensional black-box optimization tasks, with potential applications in large scale scientific machine control, robotics, and engineering design.

ACKNOWLEDGMENTS

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

## A  DISCRETE ES

Consider the following form of discrete ES step:

$$x_{i,x+1} = x_{i,t} + \sqrt{\alpha_i w_i} \cos\left(w_i t + \kappa_i f\left(\boldsymbol{x}_t\right)\right) \tag{13}$$

Then $T$ step later

$$x_{i,t+T} = x_{i,t} + \sum_{\tau=0}^{T-1} \left(x_{i,t+\tau+1} - x_{i,t+\tau}\right) \tag{14}$$

$$= x_{i,t} + \sqrt{\alpha_i w_i} \sum_{\tau=0}^{T-1} \cos\left(w_i(t+\tau) + \kappa_i f(\boldsymbol{x}_{t+\tau})\right) \tag{15}$$

Write the expansion by

$$f(\boldsymbol{x}_{t+\tau}) = f_t + \sum_{j=1}^{D} \left(x_{j,t+\tau} - x_{j,t}\right) \frac{\partial f_t}{\partial x_j} + O\left(\left(\boldsymbol{x}_{t+\tau} - \boldsymbol{x}_t\right)^2\right) \tag{16}$$

$$=: f_t + \epsilon_\tau + O\left(\left(\boldsymbol{x}_{t+\tau} - \boldsymbol{x}_t\right)^2\right) \tag{17}$$

Then

$$x_{i,t+T} = x_{i,t} + \sqrt{\alpha_i w_i} \sum_{\tau=0}^{T-1} \cos\left(w_i(t+\tau) + \kappa_i f_t\right) \tag{18}$$

$$- \sqrt{\alpha_i w_i} \sum_{\tau=0}^{T-1} \sin\left(w_i(t+\tau) + \kappa_i f_t\right) \kappa_i \epsilon_\tau + O\left(\sqrt{\alpha_i w_i}\left(\boldsymbol{x}_{t+\tau} - \boldsymbol{x}_t\right)^2\right) \tag{19}$$

The summation of the cosine term can be carried out:

$$\sum_{\tau=0}^{T-1} \cos\left(w_i(t+\tau) + \kappa_i f_t\right) = \cos\left(\kappa_i f_t + \left(t + \frac{T-1}{2}\right) w_i\right) \frac{\sin\left(\frac{T w_i}{2}\right)}{\sin\left(\frac{w_i}{2}\right)} \tag{20}$$

which is finite in the limit of $T \to \infty$.

Plugging in $\epsilon_\tau$ into the sine term of Equation 19, it reads

$$S := \sum_{\tau=0}^{T-1} \sin\left(w_i(t+\tau) + \kappa_i f_t\right) \epsilon_\tau$$

$$= \sum_{j=1}^{D} \sqrt{\alpha_j w_j} \frac{\partial f_t}{\partial x_j} \sum_{\tau=0}^{T-1} \sum_{\mu=0}^{\tau-1} \sin\left(w_i(t+\tau) + \kappa_i f_t\right) \cos\left(w_j(t+\mu) + \kappa_j f_t\right) \tag{21}$$

As $T \to \infty$, the summation can be approximated by integration

$$\lim_{T\to\infty} S = \delta_{i,j} \int_0^T d\tau \int_0^\tau d\mu \left(\sin\left(w_i(t+\tau) + \kappa_i f_t\right) \cos\left(w_i(t+\mu) + \kappa_i f(\boldsymbol{x}_t)\right)\right) \quad (22)$$

$$= \delta_{i,j} \int_0^T d\tau \frac{\sin^2\left(w_i(t+\tau) + \kappa_i f_t\right)}{w_i} = \delta_{i,j} \frac{T}{2w_i} \quad (23)$$

Therefore,

$$\lim_{T\to\infty} \frac{1}{T} \left(x_{i,t+T} - x_{i,t}\right) = -\frac{\kappa_i \alpha_i}{2} \frac{\partial}{\partial x_i} f(\boldsymbol{x}_t) \quad (24)$$

showing that the discrete ES step performs gradient descent on average. The sign before the gain parameter of the Equation 13 is chosen for minimization. Selecting an opposite sign choice will result in gradient ascent.

## B  PROOF OF THEOREM 1

Stationarity and isotropy with $\phi \in C^2$ imply, the first term of Equation 7 is

$$\nabla K(\boldsymbol{x}, \boldsymbol{x}) \nabla^\top = -\phi''(0) I_D =: \gamma I_D \quad (25)$$

where $I_D$ is an identity and $\gamma > 0$ for finite prior variance.

Write $a := \phi(0) + \sigma^2$ and $b := \phi(2r)$. Then, the kernel matrix of the second term of Equation 7 read,

$$K(\boldsymbol{X}, \boldsymbol{X}) + \sigma^2 I = \begin{bmatrix} a & b \\ b & a \end{bmatrix} =: S, \qquad S^{-1} = \frac{1}{a^2 - b^2} \begin{bmatrix} a & -b \\ -b & a \end{bmatrix}. \quad (26)$$

Thus

$$\boldsymbol{\Sigma}_g(\boldsymbol{x}) = \gamma I_D - \mathbf{A}(\boldsymbol{x}) S^{-1} \mathbf{A}(\boldsymbol{x})^\top, \quad \mathbf{A}(\boldsymbol{x}) := [\nabla_{\boldsymbol{x}} K(\boldsymbol{x}, -\boldsymbol{v}), \nabla_{\boldsymbol{x}} K(\boldsymbol{x}, +\boldsymbol{v})] \in \mathbb{R}^{D \times 2}. \quad (27)$$

At $\boldsymbol{x} = 0$, with $\alpha := \phi'(|\boldsymbol{v}|)/|\boldsymbol{v}|$

$$\mathbf{A}(0) = \begin{bmatrix} \alpha \boldsymbol{v}, & -\alpha \boldsymbol{v} \end{bmatrix} \quad (28)$$

which gives

$$\mathbf{A}(0) S^{-1} \mathbf{A}(0)^\top = \frac{2(a+b)}{a^2 - b^2} \alpha^2 \boldsymbol{v}\boldsymbol{v}^\top = \frac{2\alpha^2}{a-b} \boldsymbol{v}\boldsymbol{v}^\top, \quad (29)$$

Therefore

$$\boldsymbol{\Sigma}_g(0) = \gamma I_D - \frac{2\alpha^2}{a-b} \boldsymbol{v}\boldsymbol{v}^\top. \quad (30)$$

At $\boldsymbol{x} = \boldsymbol{v}$, with $\beta := \phi'(2|\boldsymbol{v}|)/|\boldsymbol{v}|$,

$$\mathbf{A}(\boldsymbol{v}) = \begin{bmatrix} \beta \boldsymbol{v}, & 0 \end{bmatrix}, \quad (31)$$

where we used $\phi'(0) = 0$ for the second column. Then,

$$\mathbf{A}(\boldsymbol{v}) S^{-1} \mathbf{A}(\boldsymbol{v})^\top = \frac{a\beta^2}{a^2 - b^2} \boldsymbol{v}\boldsymbol{v}^\top. \quad (32)$$

The same holds at $-\boldsymbol{v}$ by symmetry. Therefore,

$$\boldsymbol{\Sigma}_g(\pm \boldsymbol{v}) = \gamma I_D - \frac{a\beta^2}{a^2 - b^2} \boldsymbol{v}\boldsymbol{v}^\top. \quad (33)$$

Note that $\boldsymbol{\Sigma}_g(0)$ and $\boldsymbol{\Sigma}_g(\pm \boldsymbol{v})$ are form of $\gamma I_D - c \boldsymbol{v}\boldsymbol{v}^\top$ with $c \geq 0$. Following the Theorem 2.1 of (Ding & Zhou, 2007), we have

$$\det(\gamma I_D - c \boldsymbol{v}\boldsymbol{v}^\top) = \gamma^{D-1}(\gamma - c\boldsymbol{v}^2). \quad (34)$$

Therefore, followings are equivalent

$$\det \boldsymbol{\Sigma}_g(0) < \det \boldsymbol{\Sigma}_g(\pm \boldsymbol{v}) \quad \Longleftrightarrow \quad c_0 > c_{\pm \boldsymbol{v}}, \quad (35)$$

where

$$c_0 = \frac{2\,\alpha^2}{a-b}, \qquad c_{\pm v} = \frac{a\,\beta^2}{(a-b)(a+b)}.$$

(36)

The ratio is

$$\frac{c_0}{c_{\pm v}} = \frac{2(a+b)}{a} \cdot \frac{\alpha^2}{\beta^2}.$$

(37)

By assumption of nonincreasing $|\phi'(r)|$, we have $|\phi'(r)| \geq |\phi'(2r)|$, hence $\alpha^2 \geq \beta^2$. Also $a, b > 0$. Therefore $c_0/c_{\pm v} \geq 2$, which yields $\det \boldsymbol{\Sigma}_g(0) < \det \boldsymbol{\Sigma}_g(\pm v)$.

