# OpenReview forum: "Extremum Seeking with Surrogate Gradients: Scalable Derivative-Free Optimization for High-Dimensional Black-Box Functions"
_ICLR.cc/2026/Conference — ICLR 2026 Conference Withdrawn Submission_

### Official Review · Reviewer_fBPS · 2025-10-23

**Soundness:** 2
**Presentation:** 2
**Contribution:** 2
**Rating:** 2
**Confidence:** 3

**Summary:**

The paper addresses the problem of derivative-free global optimization. The proposed approach combines surrogate gradients and extremum seeking. The step estimates from both methods are combine linearly and the Adam optimizer is used on top of the surrogate gradient (or both, this part is not completely clear, see below). Additionally the authors use the gradient surrogate uncertainty as preconditioning following prior work. The evaluation compares the method with ablations thereof and with Turbo, a popular and efficient BO approach. Th results show some improvement over Turbo on 2 synthetic problems in 200d.

**Strengths:**

- The method presents improvements over Turbo on the selected baselines
- The idea is simple and intuitive. It also seems to be relatively easy to implement given a good GP implementation

**Weaknesses:**

### Contribution
- Some parts of the method seem pretty engineered like the different LR decay schedules, which raises concerns for generalizability.
- "We provide a theoretical analysis that frames ES as an efficient sampling strategy for a local surrogate model and discuss the convergence properties of our algorithm in the presence of inexact gradients." I think this is an overstatement, since the single theorem that you provide analyzes the SG step and not the ES part of your method. Therefore this should be removed as a contribution in at the end of the introduction.
- I disagree with the statement "The integration of adaptive learning (Adam) and uncertainty-aware projection schemes extends the robustness of the framework." In the conclusion. The results do not give away that uSG improves the method, second you do not evaluate the robustness of the method (robustness to what? noise?). This sentence should be modified.

### Correctness:
- As far as I can see, Eq 10 is not "the probability of ascent along a general direction v" as you claim. Nguyen et al (2022) state $P(\nabla_v f < 0) = \Phi(\frac{v^T\mu}{\sqrt{v^T \Sigma v}})$, which is not the same quantity that you describe. What you describe is the *maximum ascent* probability that the reference introduces in Theorem 3.1.

### Evaluation
- While TurBO is a good baseline, it would be nice to see a comparison with at least one more recent BO approach to be able to actually estimate the quality of the displayed improvements. Similarly, some non-synthetic baselines such as the Mujoco tasks or Rover from [1, 2] would be interesting for evaluation.
- Why do you normalize objective values by the initial values in the evaluation? This makes the results much harder to compare to previous papers. It seems like your results on 200d Ackley match those of the original TurBO paper but still, I think its bad to present this relative measure since the numbers heavily depend on the initialization then.

### Clarity
- You should define $\Delta x^{SG}$ formally before Eq (5) in your text. You do it in (6), but already refer to it before which is not nice for clarity.
- Why do you scale down $v^*$ by P? It seems like you introduce this novelty, so this should be better justified in the text.

### Minor
- Line 50 is missing a reference
- Line 77 has an awkward question mark.
- "ES not only perform gradient ascent (or descent) on average, but also introduce exploration around the optimization path." There are some words missing in this sentence.
- The dot colors are hard to tell apart in Fig 1, consider making one of them a cross or something like that.
- I think the original reference for ES should be "Ariyur, Kartik B., and Miroslav Krstić. _Real Time Optimization by Extremum Seeking Control_. Hoboken, NJ: Wiley Interscience, 2003."
- Keep the colors consisten for the methods in the paper, eg in Fig 3, keep SG orange and ES+SG in green on the right plot
- In some plots you report 10-90% quantiles (Fig 2) and in others 95% CIs (Fig 4). Choose one option and keep it consistent for clarity. In particular, mention it also in the legends of the other figures what you report.
- Eq 11 is followed by some weird repition.

---
### Sources
[1] Eriksson, David, et al. "Scalable global optimization via local Bayesian optimization." _Advances in neural information processing systems_ 32 (2019).
[2] Nguyen, Quan, et al. "Local Bayesian optimization via maximizing probability of descent." _Advances in neural information processing systems_ 35 (2022): 13190-13202.

**Questions:**

- Why do you apply Adam only to $\Delta x^{SG}$ and not to the full step including the ES update? It seems from Fig 5 that you at least decay the LRs for both
- Across how many seeds do you compute the quantiles in your eval?
- "Importantly, all SG variants remain efficient—under 5 seconds per iteration for 200D problems—making the approach practical for moderately expensive objectives while avoiding the acquisition-function overhead of BO methods." Where can I see this in the paper?

---

### Official Review · Reviewer_y4zX · 2025-10-27

**Soundness:** 2
**Presentation:** 2
**Contribution:** 2
**Rating:** 2
**Confidence:** 4

**Summary:**

The manuscript proposes an optimization method for high-dimensional Euclidean spaces in which gradient evaluations are unavailable. Although optimizing black-box functions is a critical issue to address, I have several concerns and questions. Overall, the motivation, language, and experiments need improvement.

**Strengths:**

- Scope: Optimization of high-dimensional blackbox functions has many vital applications.
- The authors propose a novel combination of methods to achieve better optimization performance in high-dimensional spaces.

**Weaknesses:**

- The motivation is unclear. Some of this impression is probably due to the writing, but it is unclear exactly how the proposed method differs from the many acquisition functions for BO. This discussion would have to cover expected improvement, probability of improvement, entropy-related functions, just to name a few.
- "However, modeling the global landscape
of the objective requires data that grows exponentially with dimensionality." I don't think this is necessarily the case for well-distributed data and customized kernels. This would, at the very least, need a citation and more explanation.
- The language has to be improved, examples:  "making single asynchronous approach relatively more trivial", " uncertainty of gradient"
- Only stationary GPs considered (esp. isotropic RBF). What about ARD kernels and simple non-stationary kernels? At least this needs a discussion.
- There are a couple of question marks that probably should not be there.
- That the posterior mean approaches the prior is very well known for GPs and can be avoided with specialized kernel designs. Again, other kernels should be considered and discussed.
-  Few and simple analytic test functions. No real function.
-  The experiment section needs to include at least various setups and acquisition functions for BO, including a gradient-exploiting BO.
-  The experiment section needs a quantitative comparison of how many iterations were needed to find the global optimum for test functions.

**Questions:**

- It is unclear to me how the proposed methodology differs from BO with a gradient-sensitive acquisition function. Where exactly can I skip a GP posterior evaluation?
- What is the performance of the algorithm compared to more suitable acquisition functions inside a BO?
- How does common anisotropy and non-stationarity change the performance of the proposed algorithms and also of the BO-based competitors?
- The actual problem with very high-dimensional spaces is that the Euclidean norm that is used in many kernels tends to contract to the same value, making learning difficult. How does your method counter that?

---

### Official Review · Reviewer_B99U · 2025-10-29

**Soundness:** 2
**Presentation:** 2
**Contribution:** 2
**Rating:** 2
**Confidence:** 4

**Summary:**

This paper proposes a black-box optimization method that combines Extremum Seeking with surrogate gradient. During optimization, the proposed ES+SG method determine the next sample direction by combining the gradient estimated by both ES and a Gaussian process surrogate model. The authors also reduce the surrogate gradient estimation variation by mid-point estimation with theoretical analysis, and accelerate the optimization process by employing Adam style learning rate adaption. Experimental results demonstrate that ES+SG with Adam outperforms one representative local Bayesian optimization baseline over three synthetic functions, and verify the contribution of algorithm components.

**Strengths:**

1. The illustration figures 1-3 help to understand the sample process of ES, as well as its advantage in estimating the surrogate gradient.

2. The experimental section ablates the algorithm components to assess their contribution.

**Weaknesses:**

1. Besides TuRBO, which was published in 2019, I think the main paper lack discussion about more recent high-dimensional BO method which is also based on local GP-based optimization, such as LA-MCTS [1], MCTS-VS [2] and MCMC-BO [3]. They and GIBO[4] mentioned in the main paper should also be consider as competitive baselines to better assess the optimization performances of ES+SG.

2. The authors only evaluate the proposed algorithm on 3 synthetic functions. I think more challenge real world benchmarks, such as those used in [1,2,3,4], can help better evaluate the algorithm performances.

3. According to Figure 6-8, combining ES+SG has marginal difference compared against SG in 2 out of 3 functions. The key component to improve the sample efficiency seems to be the Adam-based learning rate adaption. An ablation study over SG+Adam can better evaluate the algorithm design.

4. The proposed uncertainty-aware gradient step does not achieve better performances compared to the base version.

5. Some of the experimental details are unclear, which will be listed in the Question section.

6. The writing can be further improved to make the presentation clearer. (1) In e.q. (1)-(3) the authors uses $\pm$ , which may try to describe both maximization and minimization problem. A more suitable way is to only present this part under one problem setting. (2) In figure1, the color of train x and train xES are too close to be distinguishable.

7. (minor) typos in (1) line 50 and77: (?); (2) line 560: the subscript in LHS should be i, t+1?


[1] Wang, Linnan, Rodrigo Fonseca, and Yuandong Tian. "Learning search space partition for black-box optimization using monte carlo tree search." Advances in Neural Information Processing Systems 33 (2020): 19511-19522.

[2] Song, Lei, et al. "Monte carlo tree search based variable selection for high dimensional bayesian optimization." Advances in Neural Information Processing Systems 35 (2022): 28488-28501.

[3] Yi, Zeji, et al. "Improving sample efficiency of high dimensional Bayesian optimization with MCMC." 6th Annual Learning for Dynamics & Control Conference. PMLR, 2024.

[4] Müller, Sarah, Alexander von Rohr, and Sebastian Trimpe. "Local policy search with Bayesian optimization." Advances in Neural Information Processing Systems 34 (2021): 20708-20720.

**Questions:**

1. Figure 5-7 show the optimization performances in terms of iteration. Is there only one samples in each iteration? If so, I think the total evaluation number is too small in the optimization of 200D functions, which should be typically over 1000.

2. Can you show the raw values of the optimization performances instead of normalized values?

3. In line 382-384: why did you not use the default parameter setting of TuRBO?

4. The learning rate adaption seems critical to the optimization performances. Can you show an ablation study over hyperparameters in e.q. (9) ($\eta_0, \gamma, f_{ref}$)?

5. When the optimal value of the function is unknown, how to set the reference function value in e.q. (9)?

---

### Note · Authors · 2025-11-24

**Comment:**

Thank you all the valuable comments.

**Withdrawal Confirmation:**

I have read and agree with the venue's withdrawal policy on behalf of myself and my co-authors.